# Self-Confidence and Satisfaction in Simulation-Based Learning and Clinical Competence Among Undergraduate Nursing Students: A Mixed-Methods Sequential Explanatory Study

**DOI:** 10.3390/bs15070984

**Published:** 2025-07-20

**Authors:** Hadeel Anbari, Ali Kerari

**Affiliations:** 1Ministry of Health, Tabuk 47913, Saudi Arabia; hadeelanbari@gmail.com; 2Nursing Administration and Education Department, College of Nursing, King Saud University, Riyadh 11421, Saudi Arabia

**Keywords:** simulation-based learning, clinical competence, healthcare education, nursing

## Abstract

Nursing students encounter several challenges as they progress through their educational journey, particularly in integrating theoretical knowledge with practical applications using simulation-based learning (SBL). This study aimed to comprehensively assess the effects of SBL on nursing competence, self-efficacy, and overall satisfaction among undergraduate nursing students at Tabuk University, Saudi Arabia. A total of 136 students participated in this study, which employed a mixed-methods sequential explanatory design including a quantitative cross-sectional survey complemented by qualitative interviews to capture a holistic view of their experiences with SBL. The findings revealed high levels of satisfaction and self-confidence among students participating in SBL, indicating its effectiveness as an academic tool for enhancing learning outcomes. Significant positive correlations were observed among nursing competence, satisfaction, and self-confidence in the SBL context. This suggests that successful engagement in this educational approach can lead to improved clinical skills and preparedness for real-world challenges. The qualitative findings further illuminated the emotional and cognitive engagement experienced by students during the SBL sessions. The participants emphasized the importance of skill mastery in a safe and controlled environment and the positive impact of advanced technologies, such as virtual simulations, on their learning experiences.

## 1. Introduction

Healthcare education has evolved continuously to meet the demands of the dynamic healthcare sector. With the complexities of modern healthcare on the rise, there has been a significant focus on hands-on experiential learning. This trend is particularly evident in nursing, which is a field central to healthcare delivery. [12] ([12]) highlighted the need for nurses to blend theoretical knowledge with practical applications, a skillset that is increasingly essential for effective healthcare systems. In response to these evolving needs, simulation-based learning (SBL) has been introduced as a strategic solution for nursing education. SBL has been globally integrated into nursing curricula, allowing students to effectively combine their theoretical knowledge with practical experience ([15]). In clinical nursing education, SBL is not merely a trend but a necessity. It provides controlled scenarios that mirror real-life clinical situations, exposing students to various challenges and patient interactions in risk-free environments. Such engagements foster the development of critical thinking, decision-making, and practical skills, and effectively prepare students for real-world scenarios ([15]). In other words, SBL provides an interactive and dynamic learning environment that effectively bridges the gap between theoretical learning and real-world clinical practice. To fully benefit from SBL, students must develop specific learning competencies that enable them to translate simulation experiences into clinical competencies. Simulation-based learning plays a pivotal role in shaping these abilities.

Learning competence in nursing refers to the skills, knowledge, and attitudes that nursing students should acquire during their education. This plays a critical role in determining students’ ability to effectively apply these competencies in real-world clinical scenarios. Learning competence serves as a measure for assessing the efficacy of educational interventions such as SBL ([6]). The development of learning competencies is closely intertwined with the adoption of SBL because this pedagogical approach equips students with the practical skills and knowledge necessary to excel in clinical practice.

High-fidelity simulation plays a crucial role in nursing education by simultaneously supporting the development of learning competence and strengthening students’ psychological well-being. Learning competence means the skills, knowledge, and attitudes that nursing students require to deliver safe and effective care, while psychological well-being—particularly self-efficacy—reflects their confidence in performing clinical tasks ([15]; [16]). Through realistic clinical scenarios using advanced manikins, such high-fidelity simulation provides students with opportunities to practice essential skills in a safe environment; this in turn builds their self-efficacy and prepares them to handle clinical challenges with resilience and determination. By connecting competence development with increased self-confidence, high-fidelity simulation ensures that students are better prepared to provide high-quality patient care and apply what they have learned in real-world settings ([15]; [16]).

Students’ satisfaction and confidence in the learning process are crucial to nursing education. [10] ([10]) emphasized the pivotal role of simulation design and best educational practices in boosting student satisfaction and self-confidence in learning. Additionally, psychological safety serves as a crucial mediator between simulation design and student learning outcomes, thus enhancing SBL’s effectiveness ([20]).

SBL can enhance students’ overall satisfaction with education by providing them with engaging and effective learning experiences. Students who feel confident in their abilities and satisfied with their learning experiences are more likely to become competent and compassionate nurses. Psychological safety refers to students’ perception of being able to engage and learn in an environment without fear of judgment or negative consequences. When students feel psychologically safe during simulation activities, they are more likely to participate actively, ask questions, and take learning risks, which enhances satisfaction and learning outcomes ([20]). Likewise, self-confidence describes students’ belief in their ability to perform clinical tasks successfully, which directly influences their competence development ([7]).

Although research on SBL in nursing education has made significant strides, knowledge gaps need to be addressed, particularly in the context of Saudi Arabia. The adoption of SBL in Saudi Arabian nursing education signifies a commitment to producing nursing graduates who are knowledgeable, skilled, confident, and culturally competent in their clinical practice. As the healthcare landscape continues to evolve, SBL will play an integral role in preparing nurses for future challenges and opportunities.

While recent international research has increasingly focused on quantitatively measuring SBL outcomes, there remains a scarcity of qualitative studies that explore students’ in-depth experiences and perceptions, particularly in the Saudi context. This gap underscores the importance of integrating mixed methods to enrich understanding of how SBL impacts nursing competence and self-confidence. Therefore, further studies on nursing competence, learning satisfaction, and SBL among undergraduate nursing students in Saudi Arabia are required. These studies should consider the unique cultural and societal context of Saudi Arabia and explore how SBL can be tailored to meet the specific needs of the region’s nursing students. Such research may contribute to the ongoing enhancement of nursing education in Saudi Arabia and ensure that nursing graduates are well-equipped to address future healthcare challenges.

Subsequently, this study addressed the following research questions:What are the correlations between nursing competence, student satisfaction, and self-confidence in simulated learning?Are there differences in nursing competence, student satisfaction, and self-confidence in simulation learning according to the demographic characteristics of undergraduate nursing students?How do undergraduate nursing students at Tabuk University describe their experiences with SBL?

## 2. Materials and Methods

This section outlines the design, sampling, and data collection procedures used to assess the relationships among self-efficacy, nursing competence, satisfaction, and self-confidence in SBL among undergraduate nursing students at Tabuk University, Saudi Arabia. This study employed a mixed-methods sequential explanatory design, incorporating both quantitative and qualitative approaches to ensure a comprehensive understanding of the research objectives.

Tabuk University systematically integrates SBL into the undergraduate nursing curriculum as a core practical training component. Weekly high-fidelity simulation sessions are conducted in a dedicated laboratory equipped with advanced manikins that realistically replicate patient conditions. Nursing students participate in these structured sessions throughout their program. In the second year, the focus lies on basic clinical skills and patient assessment in a safe and controlled environment. In the third year, students engage in more complex simulations that require advanced clinical reasoning, critical decision-making, and teamwork. This progressive approach ensures that students gradually build their competence and confidence to apply theoretical knowledge to real-world clinical scenarios.

### 2.1. Design and Settings

This mixed-methods sequential explanatory study was conducted in two phases: a quantitative cross-sectional survey and semi-structured qualitative interviews. Data were collected at the College of Nursing, Tabuk University, where SBL has been integrated into the curriculum to enhance students’ practical skills and competencies. The quantitative phase involved an online survey distributed via Google Forms, and the qualitative phase consisted of in-depth interviews conducted virtually using Zoom. This design allowed for a thorough analysis of both numerical data and qualitative insights, thereby providing a comprehensive understanding of students’ experiences with SBL.

### 2.2. Sampling Process

This study used a two-step sampling approach. For the quantitative phase, convenience sampling targeted second- and third-year undergraduate nursing students who participated in SBL as part of their curriculum. The sample size was determined using statistical power analysis. The initial target of 114 students increased to 143 to account for potential attrition. For the qualitative phase, participants were selected from the quantitative survey based on their responses indicating significant insights or variations in SBL experiences. Fifteen students (six males and nine females) were invited to participate in semi-structured interviews, each lasting approximately 15–20 min. This purposive sampling aimed to capture diverse experiences and to provide a comprehensive understanding of SBL’s impact. Recruitment efforts included collaboration with course instructors to ensure that eligible students were informed about the study, with clear communication regarding voluntary participation and confidentiality measures.

Convenience sampling was used in the quantitative phase owing to its practicality and accessibility for reaching second- and third-year nursing students who were actively enrolled in courses with SBL sessions. This approach was deemed appropriate for exploratory research in a university setting with defined cohorts.

Triangulation was achieved by comparing and integrating quantitative findings with qualitative interview themes, allowing for a richer interpretation and validation of the study results. The instruments were reviewed by experienced bilingual nursing faculty to ensure cultural and linguistic appropriateness for the Saudi student context.

### 2.3. Data Collection Procedures

Quantitative data were collected between May and August 2024, using an anonymous online questionnaire. The questionnaire captured demographic details such as age, gender, academic level, and prior experience with SBL. In total, 146 second- and third-year nursing students completed the survey. The survey included items measuring satisfaction with the SBL experience, self-confidence in clinical skills, and perceived competence in clinical practice. The first section focused on students’ enjoyment of the simulation and the effectiveness of the teaching methods, whereas the second section explored their confidence in mastering critical content and performing clinical tasks. To ensure validity, the instrument was reviewed by experts at the SBL. It demonstrated strong internal consistency, with a Cronbach’s alpha of 0.94, indicating high reliability in assessing student satisfaction and self-confidence. Survey links were emailed to participants by instructors with follow-up reminders to maximize response rates. To mitigate potential bias, the research team ensured that instructors distributing survey links were not involved in the data analysis, and the interview questions were pre-tested for neutrality.

The qualitative phase aimed to gather in-depth insights into the students’ experiences with SBL. Fifteen students (six males and nine females) were selected for interviews based on their quantitative responses. Recruitment of nursing students was facilitated by a college’s academic advisor. Conducted via Zoom, the interviews lasted 15–20 min, and the participants were assured that their responses would remain confidential. The interviews were recorded and transcribed verbatim. Thematic analysis was used to identify key themes capturing both the positive aspects and challenges associated with SBL in nursing education. Data saturation was achieved when no new themes emerged. The research team adhered to the established qualitative research standards to ensure credibility, transferability, dependability, and confirmability. The interview questions were pre-tested with a small group of students to ensure clarity and neutrality and to enhance the quality of responses.

### 2.4. Measures

The Satisfaction and Self-Confidence Scales in SBL, developed by the National League for Nursing (NLN) and widely recognized in nursing education, were used to measure student satisfaction and self-confidence in simulation learning. This instrument was used to evaluate nursing students’ readiness to engage in clinical tasks and their satisfaction with their learning experiences ([5]). The scale consists of 13 items in total, with 5 items on the satisfaction subscale and 8 items on the self-confidence subscale. The satisfaction subscale assesses how enjoyable and effective the teaching methods and materials are, whereas the self-confidence subscale measures students’ confidence levels in mastering the content and performing clinical tasks. Both subscales employ a five-point Likert scale ranging from 1 (strongly disagree) to 5 (strongly agree). In this study, the reliability of the scale was high with a Cronbach’s alpha of 0.94, indicating strong internal consistency.

The Nursing Student Competence Scale (NSCS) was used to assess student competence in various clinical areas. The NSCS consists of 30 items covering important nursing areas and provides a comprehensive evaluation of nursing students’ competencies ([14]). The NSCS is used to evaluate several important factors among nursing students, such as medical knowledge, basic nursing skills, communication, critical thinking, lifelong learning, and global vision ([14]). All subscales are based on a five-point Likert format, with responses ranging from 1 (strongly disagree) to 5 (strongly agree). In this study, the reliability of the NSCS was high, with a Cronbach’s alpha of 0.90.

### 2.5. Data Analysis

Quantitative data were analyzed using SPSS version 27. Descriptive statistics, including means, standard deviations, and frequencies, were used to summarize demographic characteristics and scale scores. Pearson’s correlation analysis was used to assess the relationships between nursing competence, satisfaction, and self-confidence in SBL. Independent t-tests were used to examine differences in scores based on gender and academic level. Statistical significance was set at *p* < 0.05. Qualitative data were analyzed using thematic analysis following [8]’s ([8]) six-phase approach: familiarization, coding, theme generation, theme review, theme definition, and reporting. Two researchers independently coded the transcripts to ensure reliability, and discrepancies were resolved through discussion. These themes were validated against the data to ensure their accuracy and relevance to the study objectives.

### 2.6. Ethical Considerations

This study was conducted in accordance with ethical guidelines and approved by the Institutional Review Board (IRB) of King Saud University (IRB Approval Number: KSU–HE–24–235). Participation in the study was voluntary, and informed consent was obtained from all participants prior to data collection. The participants were informed of the study’s purpose, procedures involved, and their right to withdraw from the study at any time. Confidentiality was maintained by assigning unique codes to participants, and all data was stored securely with access limited to the research team.

Participation in the study was entirely voluntary and had no impact on students’ course grades or academic standing. Students were informed that declining or withdrawing from the study would not affect their relationship with faculty or their performance evaluations. Recruitment was conducted with clear assurances of confidentiality and academic independence.

## 3. Results

A total of 136 nursing students (57 males and 79 females) participated in the quantitative phase, while 15 students (6 males and 9 females) took part in the qualitative interviews. Ultimately, 6 respondents refused to participate in the online survey, and of the remaining 146 eligible participants, 10 were excluded owing to missing data or incomplete submissions. The findings addressed both quantitative and qualitative analyses, providing a comprehensive understanding of the relationship between satisfaction and self-confidence in SBL and nursing competence in clinical settings. Together, these findings offer a holistic view of the educational experience and its effectiveness in preparing students for real-world practice.

### 3.1. Sample Characteristics

Of 136 nursing students, 58.1% identified as female (*n* = 79) and 41.9% identified as male (*n* = 57). The majority of the respondents (83.4%) fell within the 19–24 age range, with 54.8% between 19 and 21 years and 39.0% between 22 and 24 years. A smaller proportion of participants were aged 25–27 and 28–30, each accounting for 1.4% of the sample, while 3.7% were aged 30–33 (see Table 1).

The academic level distributions of the 136 participants are presented in Table 1. Most (52.9%, *n* = 72) were in their third year of the Bachelor of Science in Nursing (BSN) program, while 47.1% (*n* = 64) were second-year BSN students. This near-equal representation of second- and third-year students provides a balanced view of the perceptions and experiences across different stages of nursing education.

### 3.2. Satisfaction and Self-Confidence in SBL

Table 2 shows that the students’ total satisfaction with simulation learning was high (M = 4.18, SD = 0.80). The nursing students reported the highest rating for “the effective and helpful teaching methods” used in the simulation, while the lowest rating was for the statement: The way my instructor(s) taught the simulation was suitable to the way I learn.

Regarding the nursing students’ total self-confidence in SBL, the results indicated a high level (M = 4.24, SD = 0.79) (see Table 3). The highest mean score was for the statement indicating that students needed teacher guidance on what necessary simulation activities they were required to learn during class time, while the lowest mean score was for instructors’ use of effective SBL-related resources. The findings showed no statistically significant differences between male and female nursing students regarding self-confidence (*p* > 0.05) or satisfaction with SBL (*p* > 0.05). Additionally, the results demonstrated no significant differences between second- and third-year nursing students in their levels of confidence in simulation learning (*p* > 0.05) or their levels of satisfaction with SBL (*p* > 0.05).

### 3.3. Nursing Student Competence Scale (NSCS)

The study’s findings showed that the overall mean score on the NSCS was 4.18 (SD = 0.79), indicating that the participants had a high level of nursing competence skills (see Table 4). The results also revealed that the highest average score for lifelong learning was 4.29 (SD = 0.80), while the lowest mean score was for medical-related knowledge (M = 3.98, SD = 0.52) (see Table 5). The mean scores for the other NSCS factors were basic nursing skills (M = 4.25, SD = 0.83), communication and cooperation (M = 4.21, SD = 0.89), global vision (M = 4.15, SD = 0.94), and critical thinking (M = 4.20, SD = 0.91). Accordingly, there were no significant differences in NSCS scores based on age, gender, or academic level (*p* > 0.05).

### 3.4. The Association Between the Main Study Variables

There were significant correlations between the nursing student competence, satisfaction with simulation learning, and self-confidence in SBL. Participants with higher levels of nursing competence reported greater satisfaction and self-confidence with SBL (r = 0.75, *p* < 0.001; r = 0.74, *p* < 0.001, respectively). In addition, the results demonstrated a strong positive correlation between satisfaction and self-confidence in SBL (r = 0.83, *p* < 0.001). For example, as nursing students’ satisfaction with SBL increased, their self-confidence in SBL also increased, and vice versa (see Table 5).

**Table 5 behavsci-15-00984-t005:** Correlation matrix for the main study variables.

Variable	1	2	3
1 NSCS “sum score”	1		
2 Satisfaction in SBL “sum score”	0.75 ***	1	
3 Self-Confidence in SBL “sum score”	0.74 ***	0.82 ***	1

Note: Pearson’s correlation was used. *** *p* < 0.001. NSCS, nursing student competence scale; SBL, simulation-based learning.

### 3.5. Qualitative Results

Six male and nine female nursing students participated in the qualitative analysis of the study. The participants were drawn from the second and third years, providing a diverse range of perspectives. Upon completion of the transcript analysis, five major themes emerged that captured both the positive aspects and challenges associated with SBL in nursing education.

Emotional and Cognitive Engagement in SBL. All the participants expressed strong emotional engagement during the SBL sessions. Several participants mentioned that the sessions were not only educational but also highly enjoyable, fostering a positive and supportive learning environment. The interactive nature of SBL kept students motivated and engaged throughout their learning journey, thereby enhancing their overall experience. Additionally, many participants found that the immersive quality of the simulations allowed them to stay attentive and involved in ways that traditional classroom settings do not offer. The following excerpts illustrate how students described the positive and enjoyable atmosphere created by SBL:
“I found the simulation to be a positive and enjoyable experience.”(P3)
“I felt more involved in the learning process through simulations than in traditional classroom settings.”(P2)

All participants highlighted the importance of cognitive engagement during the SBL sessions. Several participants noted that SBL’s hands-on and immersive aspects allowed them to deepen their understanding of clinical concepts and retain knowledge more effectively. The interactive nature of the simulations helped the students actively engage with the material, challenging them to apply their theoretical knowledge to real-world situations. Additionally, many participants found that SBL enabled them to link theoretical knowledge with practical applications, making it easier for them to understand complex clinical procedures. This is reflected in students’ comments about how SBL deepened their understanding and encouraged critical thinking:
“The interactive tools kept me focused and challenged me to think critically about real-world scenarios.”(P7)
“SBL helped me to link theoretical knowledge with practical applications, making it easier to understand complex clinical procedures.”(P4)

Skill Mastery and Confidence Building. All participants who participated in the SBL experience unanimously expressed that the program was instrumental in building both competence and confidence in clinical practice. Several participants mentioned that the opportunity to practice procedures in a safe, risk-free environment allowed them to refine their skills without the pressure of real clinical consequences. The opportunity to repeatedly perform tasks such as blood draws or other clinical techniques contributed greatly to their ability to confidently handle real-world scenarios. Other participants found value in practicing these tasks until they felt fully prepared for real-world applications, thereby reinforcing their competence and self-assurance. The following excerpts highlight students’ perceptions of how repeated practice improved their confidence and skill mastery:
“I was able to perform a blood draw correctly during my first time in the hospital after practicing in the simulation.”(P5)
“Repeated practice in the simulation lab gave me the confidence to handle real clinical scenarios.”(P6)

Technology’s Role in Learning. Technology emerged as a critical theme in participants’ experiences, with many students praising the use of high-fidelity mannequins and simulation software to enhance the realism of their learning environments. Several participants highlighted how advanced technological tools made their learning experiences feel more immersive and authentic. However, some students encountered challenges adapting to new technological tools, indicating that additional support may be required for certain learners. The students’ feedback below shows how they experienced the benefits and challenges of using advanced simulation technology:
“I was skeptical at first, but the detailed simulations made the learning process feel very realistic.”(P12)
“It was difficult to adjust to the software used in simulations, which was different from what we learned in class.”(P8)

The Utility of High-Fidelity Simulation. Fidelity and utility are frequently cited as essential factors of SBL effectiveness. The participants praised the high-fidelity simulations, which closely mimicked real clinical situations, for their ability to create a life-like learning environment. This realism allows students to develop and practice their clinical skills with greater confidence. They specifically noted how the advanced tools used in the simulations enhanced their learning experiences, bridging the gap between their theoretical knowledge and practical applications. However, the utility of these tools also proved to be a challenge for some, as adapting to new technologies requires additional training and support. These comments reflect how students perceived both the advantages and the limitations of high-fidelity simulation:
“The use of advanced technology significantly improved my experience by making the scenarios feel realistic.” (P3)
“It was difficult to adjust to the software used in simulations, which was different from what we learned in class.”(P6)

Suggestions for Improved Learning Outcomes. Several participants offered constructive suggestions for improving the SBL experience, emphasizing the importance of increasing diversity and cultural relevance in simulation scenarios. Some participants expressed the need for a wider range of clinical cases that better reflected the variety of situations that students might encounter in real-world practice. In addition, many felt that incorporating cultural and social contexts into simulations could greatly enhance their learning experiences. Examples of comments included the following:
“Incorporating more diverse clinical cases would provide additional learning opportunities.”(P7)
“Including scenarios that reflect the cultural and social contexts we are likely to encounter would improve learning experiences.”(P11)

## 4. Discussion

This study investigated the effectiveness of SBL in enhancing the professional competence of nursing students at Tabuk University. Specifically, it focused on key educational outcomes such as self-efficacy, clinical competence, and satisfaction with the learning process.

The results demonstrated high levels of satisfaction and self-confidence in SBL among nursing students, with mean scores of 4.23 for satisfaction and 4.29 for self-confidence. Similarly, a study conducted by [10] ([10]) assessed scores for both subscales of student confidence and learning satisfaction in simulation and found high levels of satisfaction and confidence, with scores exceeding 4.2 out of 5. Similarly, [13] ([13]) reported that participants rated their satisfaction with and self-confidence in SBL highly, with mean scores of 4.46 (SD = 0.47) and 4.44 (SD = 0.42), respectively. These findings indicate that SBL is used to enhance student engagement and preparedness with consistently high levels of student satisfaction and self-confidence across various settings ([3]).

In this study, the NSCS was used to evaluate the overall competence of the nursing students, yielding an average score of 4.18 (SD = 0.79), demonstrating a strong understanding of essential clinical skills. The highest subscale score was for lifelong learning (M = 4.29, SD = 0.80), indicating students’ recognition of the importance of continuous nursing education. The lowest score was for medical-related knowledge (M = 3.98, SD = 0.52), suggesting areas for further curriculum emphasis. These findings align with those of other studies using the NSCS, such as [14] ([14]).

Another important finding from both the Korean and Filipino studies is that there were no significant differences in satisfaction or self-confidence based on individual demographic characteristics. In a Korean study, [10] ([10]) noted no significant differences in student confidence and learning satisfaction scores based on participants’ characteristics. Similarly, [13] ([13]) found no significant differences between male and female students regarding their satisfaction and self-confidence in SBL. Furthermore, no significant differences were observed between students’ years in college and their satisfaction or self-confidence in SBL. This finding mirrors the results of the present study, in which no significant differences were found in satisfaction and self-confidence scores according to gender or academic level. This consistency across multiple studies suggests that the standardized curriculum and high-quality resources provided at Tabuk University, which are accessible to all students regardless of gender or academic level, likely contributed to uniformly high levels of satisfaction and self-confidence. It also emphasizes SBL’s role in creating equitable learning experiences for nursing students, ensuring that all participants benefit equally from this experiential learning approach, which enhances their clinical competence and preparedness for professional practice ([11]). Interestingly, there were no significant differences in NSCS scores based on demographic characteristics such as age, gender, or academic level, suggesting that the scale can be applied effectively across diverse student populations. These findings align with the idea that consistent curricula and high-quality resources ensure a uniform student experience regardless of individual differences. A study developed by [14] ([14]) assessed SBL’s impact on the clinical competencies of diverse student groups. The lack of variation in outcomes based on demographic factors supported the effectiveness of SBL in providing equitable training and improving clinical competence across different populations.

In this study, strong correlations were identified between nursing students’ self-confidence and satisfaction with simulation learning and their nursing clinical performance, as measured by the NSCS. These findings are consistent with those of previous studies using similar scales. [13] ([13]) found a positive relationship between overall satisfaction and student self-confidence. This suggests that emotional and cognitive engagement fostered by effective simulation experiences are essential for boosting nursing students’ confidence ([1]). SBL’s importance as a teaching method is further supported by findings showing that self-confidence and satisfaction with simulation learning positively correlate with students’ competence levels ([21]).

Research conducted in various international contexts has illustrated how self-confidence and satisfaction with simulation experiences directly affect students’ clinical competencies. Additionally, these studies emphasize that effective simulation designs that incorporate feedback and guide reflection are critical for fostering a supportive learning environment that enhances satisfaction and self-confidence among nursing students ([2]; [4]; [18]). Evidence from these studies underscores the necessity of implementing robust simulation strategies in nursing education to optimize student outcomes.

In addition to the quantitative findings, the qualitative data provided deeper insights into the emotional and cognitive dimensions of students’ SBL experiences. Many participants emphasized that the interactive, hands-on nature of SBL not only engaged them emotionally but also promoted cognitive growth by linking theoretical concepts with practical applications. Students noted that they felt more immersed and involved than in traditional classroom settings, with several participants highlighting the positive impact of realistic scenarios on their focus and motivation. This immersive experience allowed the students to engage more fully with the material, aiding in the retention of complex clinical knowledge and fostering a deeper understanding of critical procedures.

Furthermore, the participants expressed that the opportunity to practice skills repeatedly in a controlled, risk-free environment was instrumental in building both competence and self-confidence. They appreciated the opportunity to refine their clinical skills without the high stakes of real patient interactions, which made them feel more prepared and assured for actual clinical practice. Several students specifically mentioned that SBL boosted their confidence in performing complex procedures such as blood draws, as they had opportunities to practice until they felt proficient. These qualitative insights reinforced the quantitative findings, illustrating how SBL can enhance nursing students’ confidence and clinical competence by providing an engaging, supportive, and realistic learning environment.

These results highlight their significant impact on the nursing profession and practice. High levels of self-confidence and satisfaction with simulation learning are crucial for nursing students because these factors directly influence their ability to perform competently in clinical settings. Improved self-confidence is associated with better clinical decision-making and the ability to effectively handle complex patient situations ([17]). This is particularly relevant for success in standardized licensure examinations, such as the Saudi Nursing Licensure Exam (SNLE), where confidence in clinical skills can greatly enhance performance. SBL not only improves clinical competence and decision-making but also directly contributes to exam readiness by simulating the types of scenarios students are likely to encounter in licensure exams. By practicing in realistic, high-fidelity simulation environments, students are better equipped to manage the pressure and complexity of clinical assessments, thus improving their likelihood of success in exams such as the SNLE. By cultivating a positive learning environment through SBL, nursing education programs can prepare students for the challenges of real-world practice, ensuring that they possess the skills and confidence necessary to deliver high-quality patient care. A study developed by [9] ([9]) evaluated the impact of simulations on learning outcomes and observed a substantial average effect. The authors highlighted SBL’s significant role in equipping students with the complex skills required for success in professional exams. Thus, fostering an environment in which students experience high levels of satisfaction and self-confidence through SBL is vital for developing competent nursing professionals capable of succeeding on licensure exams and contributing positively to the healthcare system.

The findings have several practical implications and recommendations for enhancing the SBL experience at Tabuk University. Based on student feedback, incorporating more diverse and culturally relevant simulation scenarios could further enrich the learning environment, making it more reflective of the clinical diversity that students encounter in their professional careers. Additionally, integrating advanced simulation technologies such as virtual reality could provide even more immersive experiences, potentially increasing engagement and improving skill acquisition ([19]). Some students also recommended providing targeted support to those who may initially struggle with new simulation tools, which could facilitate smoother transitions and optimize learning outcomes. Future studies examining SBL’s long-term effects on clinical performance after graduation should provide valuable insights into how these skills translate in the workplace. Including students’ GPAs as a criterion in future studies could enhance the measurement of academic performance and provide a more comprehensive understanding of the factors influencing SBL outcomes. Researchers could also investigate the impact of faculty training on SBL outcomes, as instructors who are well versed in simulation-based methodologies are better equipped to facilitate meaningful learning experiences. Conducting longitudinal studies that include nursing graduates during their professional careers would enable a more comprehensive understanding of SBL’s influence over time. Additionally, expanding the research to include multiple institutions could provide comparative data, highlighting how variations in resources and curriculum structures impact SBL’s effectiveness. These insights could contribute to a more nuanced understanding of SBL and help guide future curriculum development.

In addition, the rapid development of nursing education in Saudi Arabia aligns closely with the goals of Saudi Vision 2030, which emphasizes strengthening the healthcare workforce and improving healthcare service quality. The integration of high-fidelity simulation into nursing curricula reflects this national commitment to modernize educational approaches and equip future nurses with the skills needed for increasingly complex clinical environments. Furthermore, the use of simulation-based learning represents a relatively new approach in Saudi nursing education, which has historically relied heavily on traditional teaching and clinical placements. Therefore, providing empirical evidence of its added value is essential to support its integration into existing curricula. By examining how structured simulation sessions contribute to competence and self-confidence, this study provides context-specific insights that can support ongoing educational reforms in line with Saudi Vision 2030.

Several limitations inherent to mixed methods research may have affected the interpretation and generalizability of the findings. Integrating quantitative and qualitative data poses a challenge because the two types of data may not always align seamlessly. This complicates the process of drawing cohesive conclusions. Additionally, the limited student engagement and response rates to digital surveys may have affected the quality and completeness of the data collected. Importantly, satisfaction with simulations alone is an insufficient measure of student competence, as this assessment depends on various factors, including individual differences among students and their understanding of the curriculum content. Using established frameworks for data integration and triangulation methods may help address these challenges and improve data alignment. Moreover, the extensive time and resources required to conduct both surveys and interviews often limit sample sizes, which can affect the results’ broader applicability. Qualitative data collection, particularly through interviews, may be prone to social desirability bias, in which participants provide responses believing that they are favorable or expected rather than reflecting on their true experiences. Furthermore, reliance on self-reported data introduces a potential response bias as students may overestimate their satisfaction or confidence levels. Additionally, although validated scales were used to measure competence, satisfaction, and self-confidence separately, the strong correlations observed suggest a degree of conceptual overlap that should be explored further in future research. These limitations are particularly relevant in the context of educational research, where accurately capturing student experiences and learning outcomes is crucial.

Moreover, although triangulation was conducted by integrating quantitative and qualitative data, no external triangulation or independent audit was performed, which may limit the ability to confirm the findings objectively.

## 5. Conclusions

This study underscores SBL’s crucial role in enhancing nursing students’ satisfaction, self-confidence, and clinical competence at Tabuk University. As faculty members contribute to advancing simulation labs, the nursing program provides a high-quality educational experience that remains equitable and consistent across different demographics, such as gender and academic levels. A standardized curriculum fosters inclusivity, ensuring that all students have access to the same resources and learning opportunities. As the healthcare landscape continues to evolve, particularly in response to diverse clinical demands and technological advancements, adapting SBL to include culturally relevant scenarios and incorporating innovative tools, such as virtual reality, will be essential for maintaining its efficacy and resonance with students. By fostering an environment that encourages confidence, competence, and cultural sensitivity, SBL equips future healthcare professionals with the skills and mindsets necessary for providing high-quality care in an increasingly complex and diverse healthcare system. Such commitments to continuous improvement will help support the global development of well-rounded, empathetic practitioners, enabling them to meet the evolving challenges of the healthcare industry not only in Saudi Arabia but worldwide.

In summary, this study addressed the three research questions as follows: First, the results showed strong positive correlations between nursing competence, student satisfaction, and self-confidence in simulation-based learning. Second, no significant differences were found in these variables according to demographic characteristics such as gender or academic level, indicating equitable outcomes across student groups. Third, qualitative insights confirmed that students perceived SBL as highly effective for enhancing their confidence, competence, and engagement, demonstrating the value of structured high-fidelity simulation in nursing education at Tabuk University.

## Figures and Tables

**Table 1 behavsci-15-00984-t001:** Demographic characteristics of participants (N = 136).

Variable	N	%
Age		
19 to 21	74	54.4%
22 or older	62	45.6%
Gender		
Male	57	41.9%
Female	79	58.1%
Academic Levels		
2nd year in the nursing program	64	47.1%
3rd year in the nursing program	72	52.9%

**Table 2 behavsci-15-00984-t002:** Mean scores of the items of satisfaction in simulation learning (N = 136).

Items	M	SD
1. I enjoyed how my instructor taught the simulation.	4.22	0.94
2. The way my instructor(s) taught the simulation was suitable to the way I learn.	4.07	1.06
3. The teaching methods used in this simulation were helpful and effective.	4.26	0.96
4. The teaching materials used in this simulation were motivating and helped me to learn.	4.18	1.03
5. The simulation provided me with a variety of learning materials and activities to promote my learning in different classes’ curricula.	4.18	1.05
Sum Score	4.18	0.80

**Table 3 behavsci-15-00984-t003:** Mean scores of the items of self-confidence in simulation learning (N = 136).

Items	M	SD
1. It is the instructor’s responsibility to tell me what I need to learn of the simulation activity content during class time	4.31	0.88
2. My instructors used helpful resources to teach the simulation	4.17	1.05
3. It is my responsibility as the student to learn what I need to know from this simulation activity.	4.32	0.85
4. I know how to get help when I do not understand the concepts covered in the simulation.	4.26	0.93
5. I am confident that I am mastering the content of the simulation activity that my instructors presented to me.	4.25	0.88
6. I am confident that this simulation covered critical content necessary for the mastery of medical surgical curriculum.	4.21	0.91
7. I know how to use simulation activities to learn critical aspects of these skills.	4.19	0.97
8. I am confident that I am developing the skills and obtaining the required knowledge from this simulation to perform necessary tasks in a clinical setting.	4.18	1.04
Sum score	4.24	0.71

**Table 4 behavsci-15-00984-t004:** Mean scores of the NSCS subscales (N = 136).

Subscales	M	SD
1. Medical-related knowledge	3.98	0.97
2. Basic nursing skills	4.25	0.83
3. Communication and cooperation	4.21	0.89
4. Lifelong learning	4.29	0.80
5. Global vision	4.15	0.94
6. Critical thinking	4.20	0.91
Total scores	4.18	0.79

## Data Availability

Data are not shared due to privacy and ethical restrictions.

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
