# Peer review of "Self-Confidence and Satisfaction in Simulation-Based Learning and Clinical Competence Among Undergraduate Nursing Students: A Mixed-Methods Sequential Explanatory Study"

_behavsci, 2025, doi:10.3390/bs15070984_

Round 1

Reviewer 1 Report

Comments and Suggestions for Authors

Introduction:

Line 30 - suggest changing first words from "Medical education" to either "Healthcare education" or "Nursing education"

Line 47 page 2, suggest using a transition sentence to begin the paragraph on learning competence, perhaps linking it to simulation-based learning. Same suggestion for the sentence that starts with 'Students' satisfaction and confidence' on linke 59.

Line 59 - this paragraph needs to be reworked to be sure the focus is on one concept, or ties the concepts together in a way that is cohesive.

Suggest identifying if the article will focus on high-fidelity simulation, low-fidelity simulation or standardized patients using actors.

Data collection - line 131 suggest elaborating on the issues. This seems a little out of place in this section - perhaps it should be moved to the discussion or limitations?

Suggest moving lines 191-194 up to data collection since the same information is described for the qualitative portion, it makes more sense to read this information for the quantitative portion in the same place.  Then perhaps start the results with the total of participants for the qualitative and the quantitative as listed under 'sample characteristics'.

The methods section needs much more detail on the SBL that is in the curriculum. How often is it used, how it is implemented, at what points in the education is it used, what type, what are the differences in what SBL experiences a second-year student has compared to a third year etc. Without this information, it is not clear what the impact of SBL really is. 

There is a high number of male students compared to the populations I am used to seeing.  Is there any explanation for that? I just found this interesting to see nearly half the participants were males.

Are there any connections between the themes and the results of the quantitative analysis? Anything that could be displayed in a framework or diagram/model?

Author Response

Greetings, 

Reviewer 2 Report

Comments and Suggestions for Authors

Dear Authors,

Thank you for submitting your manuscript. Please refer to the attached file for details.

I hope that my comments will contribute, even in a small way, to the improvement of your manuscript.

Best regards,

Author Response

Greetings, 

Reviewer 3 Report

Comments and Suggestions for Authors

The study addresses a timely and highly relevant topic in nursing education — the impact of simulation-based learning (SBL) on students’ clinical competence, self-confidence, and satisfaction. The mixed-methods approach and the focus on the Saudi context offer a perspective that remains underexplored in the literature.

The introduction provides an adequate review of the literature, highlighting recent international studies. The connection between self-confidence, clinical competence, and SBL is well established. However, the section could be further enriched with a more critical discussion on the scarcity of qualitative data in comparable contexts.

The methodological structure is clear and follows an explanatory sequential mixed-methods design, with well-defined quantitative and qualitative phases. The instruments used are valid and reliable (e.g., NLN and NSCS, both demonstrating high Cronbach’s alpha coefficients). Still, some improvements are suggested: justify the use of convenience sampling in the quantitative phase; describe the triangulation process between the quantitative and qualitative data more thoroughly; and clarify whether cross-cultural validation of the scales was conducted.

The results are well organized, with effective use of tables and descriptive statistics. The qualitative analysis is robust, supported by representative quotations and thematic alignment. Suggested improvements include providing interpretive summaries before qualitative excerpts and consolidating lengthy tables for better clarity.

The discussion effectively integrates the findings with existing literature, reinforcing the validity of the results. It includes valuable reflections on the role of technology, cultural diversity, and equitable access to SBL. However, a more critical analysis of methodological limitations — such as potential social desirability bias and the lack of external triangulation — is recommended.

The conclusions are well grounded, practical, and aligned with pedagogical and policy implications. They reinforce the value of SBL in developing confident, competent, and adaptable healthcare professionals.

With regard to ethics, the study was approved by an institutional review board (IRB), informed consent was obtained, and confidentiality protocols were clearly described.

The manuscript is written in clear and fluent English and demonstrates technical rigor. Nonetheless, a final language review by a native English speaker could enhance cohesion and eliminate minor repetitions.

The references are up to date, well selected and formatted according to MDPI standards. However, reference no. 7 - Braun, V. & Clarke, V. (2020). Using thematic analysis in psychology. Qualitative Research in Psychology, 3, 77–101. - has been changed, the available reference will be from 2006.

Author Response

Greetings, 

Round 2

Reviewer 1 Report

Comments and Suggestions for Authors

Thank you for the revisions which have greatly improved the article. 

There are a few minor suggestions for revisions:

Line 78- remove 'rooted in Bandura's Self Efficacy Theory' as this theory is not discussed in the article anywhere else.  The rest of the sentence and the citation are fine.

Move the next sentence that starts on line 81 up to be part of the previous paragraph.

Move the new paragraph that starts on line 97 up to start the previous paragraph so it reads: 

While recent international research has increasingly focused on quantitatively measuring SBL outcomes, there remains a scarcity of qualitative studies that explore students’ in-depth experiences and perceptions, particularly in the Saudi context. This gap under-
scores the importance of integrating mixed methods to enrich understanding of how SBL 
impacts nursing competence and self-confidence. Therefore, further studies on nursing competence, learning satisfaction, and SBL among undergraduate nursing students in Saudi Arabia are required. These studies should consider the unique cultural and societal context of Saudi Arabia and explore how SBL can be tailored to meet the specific needs of the region’s nursing students. Such research may contribute to the ongoing enhancement of nursing education in Saudi Arabia and ensure that nursing graduates are well-equipped to address future healthcare challenges.

Author Response

Greetings, 
